# Delayed Leaf Senescence Improves Radiation Use Efficiency and Explains Yield Advantage of Large Panicle-Type Hybrid Rice

**DOI:** 10.3390/plants12234063

**Published:** 2023-12-03

**Authors:** Jun Deng, Tian Sheng, Xuefen Zhong, Jiayu Ye, Chunhu Wang, Liying Huang, Xiaohai Tian, Ke Liu, Yunbo Zhang

**Affiliations:** 1MARA Key Laboratory of Sustainable Crop Production in the Middle Reaches of the Yangtze River, College of Agriculture, Yangtze University, Jingzhou 434025, China; 15926599893@163.com (J.D.); atianer2558@163.com (T.S.); yejiayu990909@163.com (J.Y.); 201772394@yangtzeu.edu.cn (C.W.); lyhuang8901@126.com (L.H.); xiaohait@sina.com (X.T.); ke.liu@utas.edu.au (K.L.); 2Agricultural and Rural Bureau of Duodao District, Jingmen 448000, China; zhongxuefen198902@163.com

**Keywords:** yield, panicle-type, hybrid rice, leaf senescence, radiation use efficiency, dry matter accumulation

## Abstract

Super hybrid rice with predominantly large panicle types has achieved remarkable success in enhancing crop yield. However, when compared with multi-panicle-type varieties, the yield stability of large panicle-type varieties remains a challenge, and limited information is available on the comparative advantages of multi-panicle types. Consequently, a two-year experiment was conducted to evaluate the grain yield, biomass production, leaf area index (LAI), and radiation use efficiency (RUE) of large panicle-type hybrid rice (Y-liangyou 900, YLY900) and multi-panicle-type hybrid rice (C-liangyouhuazhan, CLYHZ) under three nitrogen (N) treatments (0, 180, 270 kg N ha^−1^). The effects of increased N fertilization were more pronounced in the large panicle-type varieties. YLY900 outperformed CLYHZ in terms of average yield (6% higher), and its yield advantage was attributed to higher spikelets per panicle (28%). Due to YLY900’s RUE being 9% higher than CLYHZ, it results in a 12% greater accumulation of dry matter than CLYHZ. Furthermore, YLY900 exhibited significant improvements of 16%, 4%, and 14% in specific leaf weight, effective leaf area ratio, and LAI at 20 days after the heading stage (20DAH), respectively, compared with CLYHZ. YLY900 also demonstrated a stronger correlation between rice yield and intercepted photosynthetically active radiation (IPAR) compared with CLYHZ, with R^2^ values of 0.80 and 0.66, respectively. These findings highlight the superior performance of YLY900, resulting from higher light interception percentage (IP) and IPAR values, which consequently led to enhanced RUE and grain yield. Our research reveals that delayed leaf senescence by increasing LAI at the post-heading stage for large panicle-type hybrid rice, thereby contributing to greater RUE, led to higher biomass production and grain yield.

## 1. Introduction

Rice occupies a prominent position among the world’s vital food crops, serving as the staple sustenance for 65% of the population [1]. The proliferation of urbanization and rapid economic progress has led to a surge in global population, giving rise to a decline in arable land per individual and intensifying the demand for sustenance [2]. Climate change, in recent years, has also ushered in extreme climatic phenomena such as droughts and floods, thereby gravely impacting the long-term stability of agricultural output [3,4,5]. It is predicted that by 2030, the world’s rice consumption will have increased from 480 million tons of milled rice in 2014 to around 550 million tons [6]. Enhancing grain productivity per unit area has thus taken on tremendous significance. The number of spikelets adorning each panicle emerges as a pivotal factor influencing yield. An in-depth exploration of the growth and developmental characteristics pertaining to diverse panicle-type rice varieties is therefore essential, as it promises an amplified rice yield and upholds food security.

Varieties classified as having a panicle of substantial size exhibit a smaller number of panicles per plant, a greater number of spikelets per panicle, and a significant contribution of spikelets per panicle to overall yield. Conversely, in varieties categorized as multi-panicle, there is a relatively high number of panicles per plant but a lower number of spikelets per panicle, resulting in a relatively higher contribution of the panicles per unit area to overall yield [7].

The notion has been put forth that enhancing the tillering ability and augmenting the count of productive panicles constitutes a pathway towards achieving high and consistent yields [8]. Multi-panicle varieties with robust tillering capacity and uniform grain-filling maturation are well-suited for mechanized, low-intensity cultivation [9]. Nevertheless, the proficient tillering capacity of multi-panicle varieties engenders issues such as excessive density, resulting in stunted individual growth, inadequate field ventilation, and insufficient light penetration. This, in turn, renders them susceptible to pests and diseases, as well as prone to lodging and premature senescence [10]. Consequently, while multi-panicle rice varieties exhibit commendable stability in terms of yield, their potential for further yield enhancement remains constrained.

In recent years, there has been a prevailing trend in breeding to appropriately diminish the quantity of panicles, considerably augment the weight of panicles, and selectively cultivate varieties with magnificently large panicles to attain a bountiful yield [11]. Numerous studies have indicated that large-panicle rice holds a greater potential for yield compared with rice with multiple panicles [12,13]. Notably, large-panicle varieties exhibit the advantage of amassing dry matter, particularly following the blossoming phase [14]. An abundant post-flowering dry matter accumulation is crucial for achieving high yields in large-panicle varieties [15]. These varieties showcase characteristics such as an extended grain-filling period, gradual leaf senescence, elevated leaf area index, a protracted functional period of effective leaf area, and formidable photosynthetic capacity in post-flowering leaves, thereby facilitating the accumulation of an amplified amount of post-flowering dry matter [14,16].

Currently, many large panicle and oversized large panicle-type varieties have been developed, such as the International Rice Research Institute’s (IRRI) new plant-type rice and China’s super rice varieties. [17]. In 2000, Liangyoupeijiu achieved a grain yield of 10.5 t ha^−1^; in 2004, Y-liangyou 1, achieved a grain yield of 12.0 t ha^−1^; in 2011, Y-liangyou 2 reached a grain yield of 13.9 t ha^−1^; in 2014, Y-liangyou 900 (YLY900) created a yield record of 15.4 t ha^−1^; in 2018, Chaoyou1000 broke the yield record of 17 t ha^−1^ [18]. By 2021, China had 135 rice varieties that met the super rice standard [19]. Super hybrid rice varieties showed yield potential and made an important contribution to food security.

However, the challenges associated with the cultivation of extensive areas with large panicle-type and oversized panicle-type rice varieties are gradually becoming apparent. For instance, the utilization of YLY900 and Chaoyou1000 varieties for a yield target of 15.0 t ha^−1^ in Yunnan revealed certain issues [20]. These varieties exhibited over 300 spikelets per panicle, as well as robust stalks and large panicle types. However, they displayed weak tillering ability and a lower number of effective panicles. To achieve high yields, extensive amounts of nitrogen fertilizer, up to 420 kg ha^−1^, must be applied—nearly twice the amount used by the average farmer in a single season [21]. With the increasing yield of super hybrid rice, there has been a corresponding surge in demand for fertilizers, resulting in reduced economic efficiency and an elevated risk of ecological pollution [22]. In addition, it is worth mentioning that super hybrid rice exhibits notable ecological adaptations, while the yields of these remarkable rice varieties are prone to environmental influences [23]. A study revealed a substantial discrepancy in yields between regional trials of super rice varieties and their high yield records, typically ranging from 3.6 to 6 t ha^−1^ [24]. This variance primarily arises from the vulnerability of glume fertilization in rice varieties with large panicles to environmental factors. The flowering stage is particularly sensitive to elevated temperatures, and heat stress during this period results in diminished pollen viability and reduced glume fertilization [25,26]. Hence, the yield performance of large panicle-type varieties varies significantly among different growing seasons and ecological regions, rendering them unstable and severely limiting their applicability in production.

At present, there is little information to compare the dry matter accumulation, radiation use efficiency, and yield of different panicle-type rice varieties. In this experiment, we compared the dry matter accumulation, radiation use efficiency, and yield of two different panicle types of rice varieties. The study aimed to (1) compare grain yield and yield components of large panicle-type and multi-panicle-type varieties under different N rates; (2) identify dry matter accumulation and radiation use efficiency under different N rates; and (3) establish the relationship between grain yield and RUE in different panicle-type varieties.

## 2. Results

### 2.1. Weather Conditions

The recorded data for daily maximum temperatures reached an identical value of 37 °C for both examined years, whereas the daily minimum temperatures slightly differed, with values of 16 °C and 17 °C in 2020 and 2021, respectively (Figure 1). During the rice growth period in 2020, the average temperature was 26.2 °C, and the total precipitation was 816.5 mm. In 2021, the average temperature during the rice growth period was 25.4 °C, with a total precipitation of 1103.4 mm. In addition, the cumulative solar radiation during the rice growing season in 2021 amounted to 2016.3 MJ m^−2^, surpassing the observed radiation in 2020 by a 15% margin.

### 2.2. Grain Yield

Increasing N significantly improved the grain yields, with a significant difference between the two rice varieties (*p* < 0.05), and the trend was consistent across the year (Figure 2 and Table 1). The average yield of YLY900 was 10.8 t ha^−1^ across two years, 6% higher than that of CLYHZ. YLY900 and CLYHZ both reached the highest yields under N3 treatment, with yields of 11.9 t ha^−1^ and 11.2 t ha^−1^ across two years, respectively. Compared with N1, the average annual yield of YLY900 increased by 23% under N2 and 29% under N3 treatment, while CLYHZ increased by 18% and 25%, respectively.

### 2.3. Yield Components

Across N treatments and varieties, there were substantial differences in panicle quantity, spikelets per panicle, and grain filling (Table 1 and Table 2). The panicle number of CLYHZ greatly increased due to increased N supply, with N3 being 31% and N2 being 21% higher than N1, respectively. Spikelets per panicle of YLY900 were dramatically increased by increased N supply, with N3 being 29% and N2 being 18% higher than N1, respectively. The spikelets per panicle of YLY900 were 28% higher than that of CLYHZ, but CLYHZ produced 44–51% higher panicle numbers than YLY900.

Grain filling decreased due to increased N supply in both CLYHZ and YLY900. The grain weights of the two rice types did not significantly increase with increased N supply; however, there were differences between the varieties, with YLY900 typically having a larger grain weight than CLYHZ.

### 2.4. Leaf Area Index, Specific Leaf Weight, and Effective Leaf Area Ratio

Throughout the course of two years, N significantly (*p* < 0.05) affected the LAI of the two rice types at different stages of growth (Figure 3). The LAI of CLYHZ was higher than that of YLY900 during MT to PI, but the difference was not significant. At HD, the LAI of CLYHZ under N1 and N2 treatments was higher than that of YLY900 but lower than YLY900 under N3 treatment. At 20DAH, the LAI of YLY900 was higher than that of CLYHZ under three nitrogen treatments. Moreover, the average LAI of YLY900 was 2.9, 14% higher than that of CLYHZ.

Increased N supply significantly improved the SLW of the two rice varieties (Figure 4). Among them, both YLY900 and CLYHZ obtained higher SLW under N2 treatment, with 67.6 g m^−2^ and 58.4 g m^−2^, respectively. Compared with the N1 treatment, the average SLW of YLY900 increased by 33% in N2, while CLYHZ increased by 19%, respectively. In addition, the average SLW of YLY900 was 12% higher than that of CLYHZ across three nitrogen treatments.

Increased N significantly increased the effective leaf area ratio in the two rice varieties (Figure 4). The average effective leaf area ratio of YLY 900 under three N treatments was 63%, 10% higher than that of CLYHZ. In addition, compared with the N1 treatment, the average effective leaf area ratio of YLY900 increased by 21% under the N2 treatment and 26.7% under the N3 treatment, while CLYHZ increased by 8% and 23%, respectively.

### 2.5. IR, IP, IPAR, and RUE

The TDW at MA in the two rice varieties was dramatically raised by increased N supply (Table 1 and Table 3). Relative to the N1 treatment, the average TDW of YLY900 was 23% higher in the N2 treatment and 33% higher in the N3 treatment, while CLYHZ had increased by 33% and 32%, respectively. YLY900 produced 11.8% higher TDW than CLYHZ over three N treatments. The TDW of YLY900 at the post-heading stage demonstrated a 17.9% increase in the N1 treatment, a 11.6% increase in the N2 treatment, and a 16.0% increase in the N3 treatment when compared with CLYHZ.

Although RUE was not significantly different between the N2 and N3 treatments, increasing N supply did promote IPAR, IP, and RUE in the rice types. YLY900 outperformed CLYHZ in the N1 and N3 treatments in terms of RUE. In the N1 treatment, the mean RUE of YLY900 was 23% higher than that in CLYHZ, and in the N3 treatment, it was 15% higher. Across variations, there were no discernible changes in growth times.

### 2.6. Grain Yield and Yield Component Analysis

Instead of grain weight and yield, a significant connection was discovered between panicle quantity, spikelets per panicle, and grain filling (Figure 5). Among them, the panicle number and spikelets per panicle of YLY900 were highly correlated with grain yield, with R^2^ values of 0.84 and 0.62, respectively. The grain yield of CLYHZ had a higher correlation between panicle number and spikelets per panicle, with R^2^ values of 0.77 and 0.66, respectively.

### 2.7. Relationship between IP, IPAR, TDW, RUE, and Grain Yield

Across varieties, IP, IPAR, TDW, and RUE were all significantly (*p* < 0.01) positively linked with grain yield. (Figure 6). For YLY900, R^2^ = 0.86 was the coefficient of determination between grain yield and TDW (*p* < 0.01), and R^2^ = 0.74 for CLYHZ. The YLY900’s IPAR and grain yield correlation coefficients were R^2^ = 0.80 (*p* < 0.01) and R^2^ = 0.63 (*p* < 0.01) for CLYHZ. The correlation coefficients of grain yield with RUE and IP for YLY900 were R^2^ = 0.77 and R^2^ = 0.74, respectively, and R^2^ = 0.42 and R^2^ = 0.65 for CLYHZ.

## 3. Discussion

Large panicles and high sink capacity are important traits in super hybrid rice breeding [11]. Previously, many studies showed that a significant increase in sink capacity by increasing the panicle number as well as the spikelets per panicle is the main factor in increasing yield [17,23,27]. The key to achieving high yields in super rice is to significantly increase the spikelets per panicle [28,29]. The varieties planted on a large scale in the 1990s (before the super rice breeding program was proposed) were medium panicle-type varieties, with relatively few large panicle-type varieties. The total spikelets of these varieties are around 3 × 10^8^ ha^−1^ and the yield is 7.5–9 t ha^−1^. In 1996, the total spikelets of Liangyoupeijiu were 4.5 × 10^8^ ha^−1^; then, Y-liangyou 1 and Y-liangyou 2 reached 5.25 × 10^8^ ha^−1^ total spikelets; the total spikelets of YLY900 and Chaoyou1000 gained 9 × 10^8^ ha^−1^ total spikelets, and their spikelets per panicle reached 300–390 [30,31].

In this study, the large-panicle rice variety YLY900 and the multi-panicle rice variety CLYHZ were chosen and tested under three different nitrogen fertilizer conditions. The results found the yield of YLY900 was 6% higher than for CLYHZ. This demonstrated that the large panicle-type varieties have great potential to obtain a high yield, consistent with the findings of previous studies [12,13]. Both the panicle number and the spikelets per panicle grew significantly with nitrogen fertilizer in both YLY900 and CLYHZ; however, the panicle number increased more in CLYHZ and the spikelets per panicle increased more in YLY900. The spikelets per panicle of YLY900 increased by 28% more than CLYHZ. The yield of YLY900 was mainly increased by increasing the spikelets per panicle. In addition, in comparison to CLYHZ, although the grain filling of YLY900 is not dominant, its 1000-grain weight was always higher than that of CLYHZ. This may be an important factor in why the yield of large panicle-type varieties is higher than the yield of multi-panicle-type varieties. Varieties exhibiting multiple panicles, like CLYHZ, exhibited a profusion of panicles, thereby amplifying the overall pollen yield per plant and elevating the grain setting rate [10,12]. However, the augmented number of panicles creates a relative deficit in the distribution of essential nutrients, moisture, and light energy. This can lead to imbalances in nutrient allocation, competitive disadvantages, and decreased resource utilization efficiency. In contrast, varieties possessing larger panicles, like YLY900, had fewer panicles, enabling the concentration of resources (nutrients, moisture, and light energy) within each panicle [8,12]. This concentrated allocation trait facilitates the enhancement of nutrient and moisture utilization efficiency while also bolstering the development and yield performance of each individual panicle. Therefore, increasing spikelets per panicle of large panicle-type varieties and ensuring a certain grain filling are important factors.

In addition, it has been suggested that a large increase in the total spikelets can lead to insufficient filling and an increase in weak grains, resulting in a reduction in yield due to a decrease in grain filling [32]. Previous studies showed that the yield performance of super rice varies greatly in different regions, mainly due to the differences in climatic conditions that have a large impact on grain filling [23,33]. Therefore, it is important to breed varieties with environmental tolerance or adjust the sowing period to ensure the grain filling of large panicle-type rice varieties for a high and stable yield [34,35]. In our study, the small interannual variation in rice grain filling between the two varieties may be due to the lack of significant differences in climatic conditions between the two years.

Total biomass and harvest index determine crop yield [36]. Although increasing biomass or harvest index can serve to increase grain yield, it was found that significantly increasing biomass is the material basis for further increasing grain yield. A previous study of the yield level of 15 t ha^−1^ in Yunnan found that the increase in the daily dry matter accumulation per unit area obtained a higher grain yield [37]. Super high-yield rice varieties have high dry matter accumulation, particularly during the middle and late growth stages [23]. In this study, the highest dry matter accumulation of YLY900 was under N3 treatment, which was 15% higher than that of CLYHZ. These results demonstrate that the biomass of large panicle-type rice varieties significantly increases under high nitrogen conditions, which is conducive to high-yield formation. Notably, there was a 15.0% increase in the accumulation of dry matter after flowering in YLY900 compared with CLYHZ. This could be considered a significant factor contributing to the yield advantage of YLY900.

Making improvements to grain filling and the appropriate source-sink relationship is the key to a high yield of large panicle-type rice [32]. To improve source supply, leaf area can be increased, or the photosynthetic capacity of leaves can be improved. Maintaining higher leaf photosynthesis after the heading stage could improve biomass accumulation in rice, as photosynthesis contributes 60% to 80% of rice grain yield after heading [38,39]. It has been discovered that the leaf area index of CLYHZ is higher than that of YLY900 with low nitrogen treatment but significantly lower with high nitrogen treatment in the early stage of rice growth. This may be due to the higher tillering ability and larger total green leaf area of multi-panicle type rice varieties in low nitrogen treatment situations, which improve the photosynthetic leaf area index. In this study, LAI of CLYHZ decreased under N3 treatment, whereas YLY900 increased. This proves that large panicle-type varieties have a greater requirement for nitrogen fertilizer than multi-panicle-type varieties.

High nitrogen is beneficial to the improvement of the tillering ability of YLY900 and the increase and thickening of leaves, thereby improving the LAI, SLW, and efficient leaf area ratio. This is conducive to the accumulation and transport of dry matter. In addition, the LAI of both varieties decreased as the growth period developed, while the LAI of YLY900 was found to be significantly higher than CLYHZ. This may be due to the leaf senescence of CLYHZ being aggravated in the heading stage, while the YLY900 was delayed because of its greater nitrogen absorption capacity. This led to both having a higher photosynclous leaf area during the later growth stage [32,38]. Therefore, increasing the LAI at post-heading and delaying leaf senescence is conducive to the accumulation of dry matter.

By raising both the photochemical vegetation index and the amount of chlorophyll in the canopy spectrum, nitrogen raises RUE. Some studies have shown that the high yield potential of hybrid rice varieties obtained through an increased nitrogen application rate is the result of increasing aboveground biomass by RUE [37,40]. Yield potential is dependent on biomass under RUE and high nitrogen treatment [41]. The main method for affecting the biomass of hybrid rice is prolonging the growth period. Increased RUE in YLY900 may be connected to leaves’ improved photosynthetic efficiency [42]. Higher RUE is related to the improvement of the photosynthetic characteristics of leaves [40]. The high-yield potential of high nitrogen regulation is a result of higher RUE transforming more biomass [40,43]. The biomass of super hybrid rice with increased nitrogen application is mainly achieved by extending the growth period and green leaf duration as a means of achieving an IPAR increase [37,44]. In this study, YLY900 achieved the highest RUE under the N3 treatment, and the RUE of multi-panicle-type varieties achieved the highest under the N2 treatment. Analysis of the causes revealed that there was no significant difference in the IP and IPAR of the two rice varieties, but the difference in their TDW was significant. These results showed that the high yield of YLY900 was mainly due to its higher RUE under the high nitrogen treatment, which led to more biomass, consistent with the previous studies [40,44].

## 4. Materials and Methods

### 4.1. Experimental Site and Test Material

A two-year field experiment was conducted in 2020 and 2021 at the experimental farm of Yangtze University (112°31′ E, 30°21′ N) in Jingzhou, Hubei Province, China. Prior to commencing the experiment each year, soil samples were procured from the top 20 cm of soil. Soil samples were taken from each treatment’s center and four corners to evaluate the characteristics of the soil. The values obtained from the four soil samples were then averaged to establish the yearly soil characteristic value. Alluvial calcareous soil is the type of soil found at the site. Average values over a two-year period show that the soil has a pH of 6.7, an organic matter content of 18.0 g kg^−1^, an al-ka-li-hydrolysable nitrogen content of 115.5 mg kg^−1^, readily available phosphorus content of 22.7 mg kg^−1^, and available potassium content of 111.2 mg kg^−1^.

In the experiment, Y-liangyou 900 (YLY900) and C-liangyouhuazhan (CLYHZ) were used as test materials. YLY900 is a new super hybrid rice variety developed from Y58S as a mother and R900 as a father. CLYHZ is a new indica-type, two-line hybrid rice cultivar. It was chosen using C815S as the mother and Huazhan as the father. The experimental materials were two hybrid rice cultivars that are commonly farmed in China.

### 4.2. Experimental Design and Crop Management

The experiments were conducted using a split-plot design, where the main plots consisted of three nitrogen treatments (N1: 0 kg ha^−1^, N2: 180 kg ha^−1^, and N3: 270 kg ha^−1^), and the subplots consisted of the varieties YLY900 (YLY900) and C-liangyouhuazhan (CLYHZ). Each study year included three replicates, and each plot had a size of 30 square meters. Urea was applied at 50%, 20%, 20%, and 10% during the transplanting, middle tillering (MT), panicle initiation (PI), and heading (HD) stages, respectively. For the K fertilizer treatments, KCl was used in a split form (rate: 100 kg K_2_O ha^−1^). Half was applied as a basal dressing, and the other half was broadcast at PI. Phosphorus (P) was broadcast as a basal fertilizer in the form of calcium superphosphate at a rate of 40 kg P_2_O_5_ ha^−1^.

Pre-germinated seeds were sown in a seed bed at 25 g m^−2^, and seedlings were transplanted to field plots at 30- to 32-several days old, with hill spacings of 20 cm × 23.3 cm and two seedlings per hill. The fields were submerged in water for 5 days after transplantation and then continuously immersed in shallow water until 5 days prior to the PI. After that, the fields were drained for 5 d, irrigated during PI, and subsequently flooded until the onset of flowering. The combined approach of shallow water depth with wetting and drying (SWD) was adopted for all treatments. In treatments using SWD water management, the fields were irrigated to a depth of 3.0 cm, allowed to dry, and then reirrigated to the same depth before any visible cracks appeared on the soil surface. Insect and disease infestations were controlled chemically throughout the entire crop growth cycle. Crop management adhered to accepted cultural practices.

### 4.3. Sampling and Measurements

Plants from six hills were sampled at the MT, PI, HD, and maturity (MA) stages. Straws, leaves, and panicles (at the HD and MA stages) were isolated from the plant samples. The leaves of each treatment were spread out flat and pasted on the whiteboard (with a 1 cm scale left on the whiteboard). Then, the camera was used to take photos 1 m away from the whiteboard each time, and the software ImageJ (v1.8.0) was used to process the photos and calculate the leaf area, leaf area index (LAI), and effective leaf area ratio (100× the leaf area of the top three leaves/the total leaf area). After that, straw, leaves, and panicles were oven-dried at 70 °C to a constant weight for dry weight determination. The specific leaf weight (SLW) in the HD stage was computed by adding the dry weight of the leaves to the HD stage. Plants from six hills were sampled diagonally from a 5 m^2^ harvest area to determine the yield components at the MA stage and the total dry weight of aboveground sections (TDW). To find the number of panicles per square meter, the number of panicles per hill was computed. Straws, leaves, and panicles were the three divisions of plants. Unfilled spikelets were isolated from hand-threshed panicles by immersing them in tap water. Three 30 g filled spikelet subsamples and three 3 g unfilled spikelet subsamples were analyzed in order to count the spikelets. The dry weights of the full and empty spikelets, as well as the rachis, were measured after being oven-dried to a consistent weight at 70 °C. The above-ground TDW was computed using the total dry matter of the straw, rachis, and filled and unfilled spikelets. Grain filling % (100 filled spikelet number/total spikelet number) and the number of spikelets per panicle were calculated. Grain yield was calculated using a 5 m^2^ plot area and adjusted to 0.14 g H_2_O g^−1^ of moisture.

### 4.4. Measurement of IR and RUE

The SunScan Canopy Analysis System was used to measure canopy light interception between 11.00 and 13.00 h at MT, PI, HD, and MA (Delta-T Devices Ltd., Burwell, Cambridge, UK). Each plot’s canopy light intensity was measured by setting the light bar halfway between two rows and just over the water’s surface. Both three readings between rows and three readings within rows were taken. At the same time, the amount of incoming light was measured. Calculated as [100× (incoming light intensity-light intensity inside the canopy)/incoming light intensity], this is the intercepted percentage of incoming light intensity by the canopy. It was discovered that 0.45% of the total solar energy was intercepted by photosynthetically active radiation, or IPAR [45].

The IPAR during each development stage was calculated using the average canopy light absorption and the total solar radiation absorbed throughout the stage [1/2 (beginning of the development stage canopy light interception and ending of the growth stage canopy light interception) cumulative radiation exposure during the growth phase]. To calculate the IPAR for the entire growing season, the intercepted radiation from each growth stage was added. The RUE was calculated using the above-ground TDW to IPAR ratio for the entire growing season. A Vantage Pro2 weather station (Davis Instruments Corp., Hayward, CA, USA) was used to record the sun radiation and lowest and maximum temperatures every day.

### 4.5. Data Analysis

The genotypic means were compared using least significant differences (LSD) at a significance threshold of 0.05, unless otherwise indicated. The data were analyzed using analysis of variance (Statistics v27.0.1, Analytical Software, Tallahassee, FL, USA).

## 5. Conclusions

In this study, we investigated the TDW, RUE, LAI, and yield components of two panicle-type rice varieties under three different N treatments over two years. We discovered that, compared with multi-panicle-type rice, (1) the large panicle-type rice has a higher yield advantage, attributed to its superior spikelets per panicle; (2) the large panicle-type rice has better RUE in high nitrogen conditions, improving its ability to accumulate dry matter; and (3) delaying leaf senescence in large panicle-type rice can increase the post-heading LAI, enhancing biomass accumulation and increasing grain yield through better RUE.

## Figures and Tables

**Figure 1 plants-12-04063-f001:**
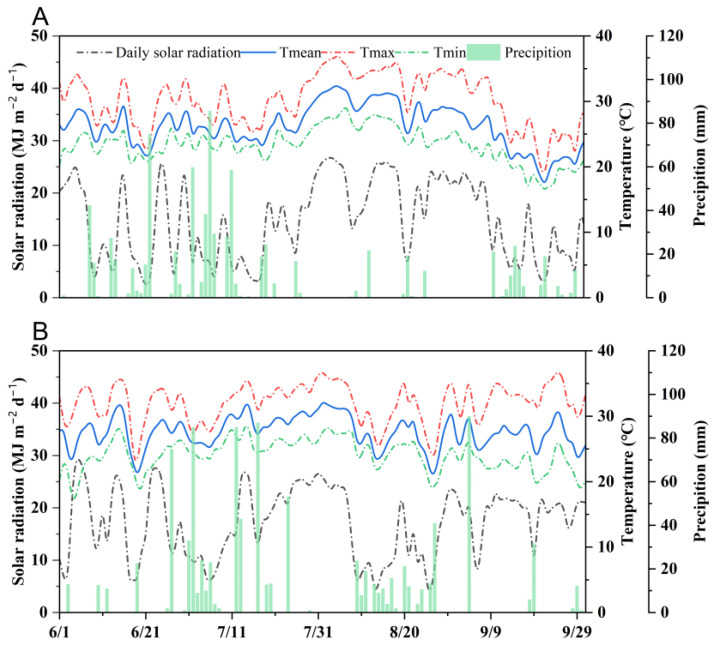
Daily maximum temperature, daily minimum temperature, and daily solar radiation during the rice-growing season at Jingzhou in 2020 (**A**) and 2021 (**B**).

**Figure 2 plants-12-04063-f002:**
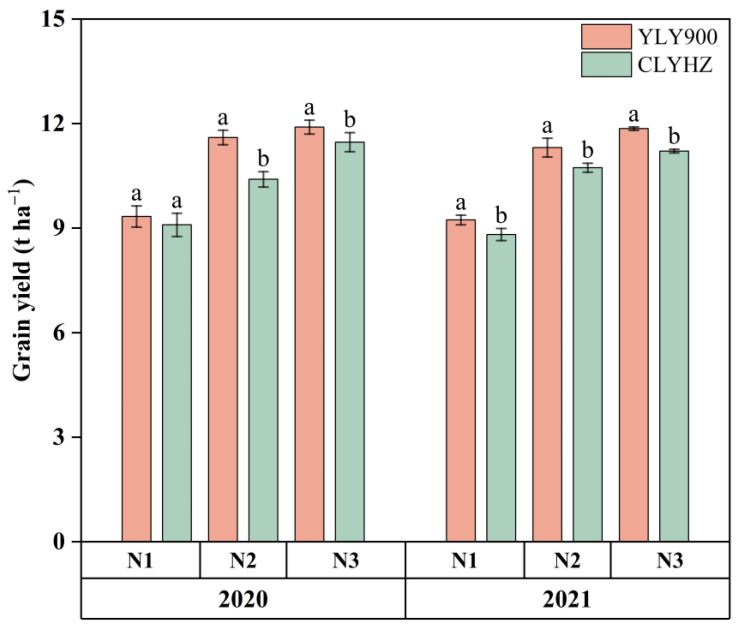
Grain yield of two panicle-type varieties under different N rates at Jingzhou in 2020 and 2021. Vertical bars indicate standard errors (*n* = 3). Means with identical letters within each column do not exhibit statistically significant differences according to LSD (0.05).

**Figure 3 plants-12-04063-f003:**
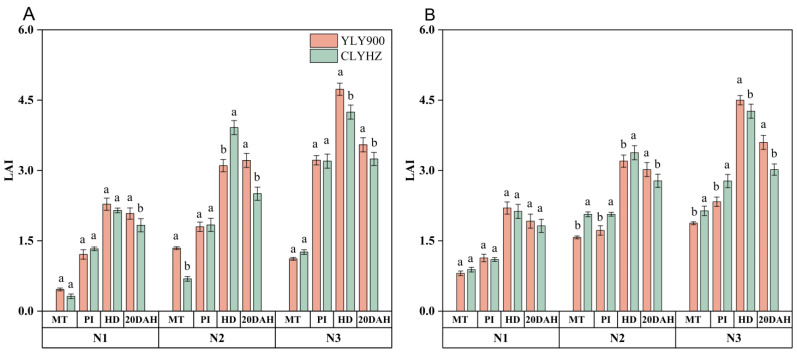
The leaf area index (LAI) of two panicle-type varieties under different N rates at Jingzhou in 2020 (**A**) and 2021 (**B**). Vertical bars indicate standard errors (*n* = 3). MT: middle tillering, PI: panicle initiation, HD: heading stage, and 20 DAH: 20 days after heading stage. Means with identical letters within each column do not exhibit statistically significant differences according to LSD (0.05).

**Figure 4 plants-12-04063-f004:**
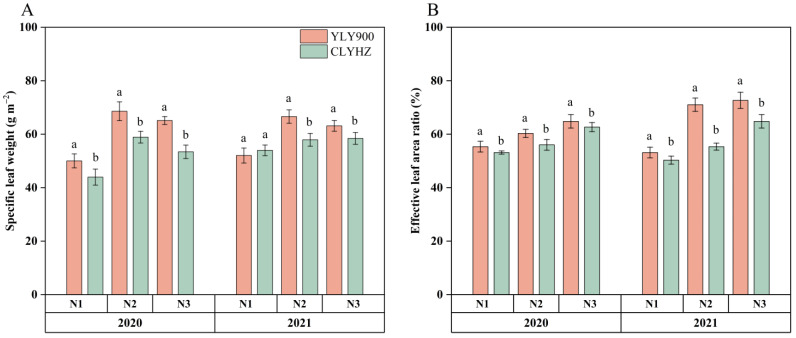
The specific leaf weight (SLW) (**A**) and effective leaf area ratio (**B**) of two panicle-type varieties during the heading stage under different N rates at Jingzhou in 2020 and 2021. Vertical bars indicate standard errors (*n* = 3). Means with identical letters within each column do not exhibit statistically significant differences according to LSD (0.05).

**Figure 5 plants-12-04063-f005:**
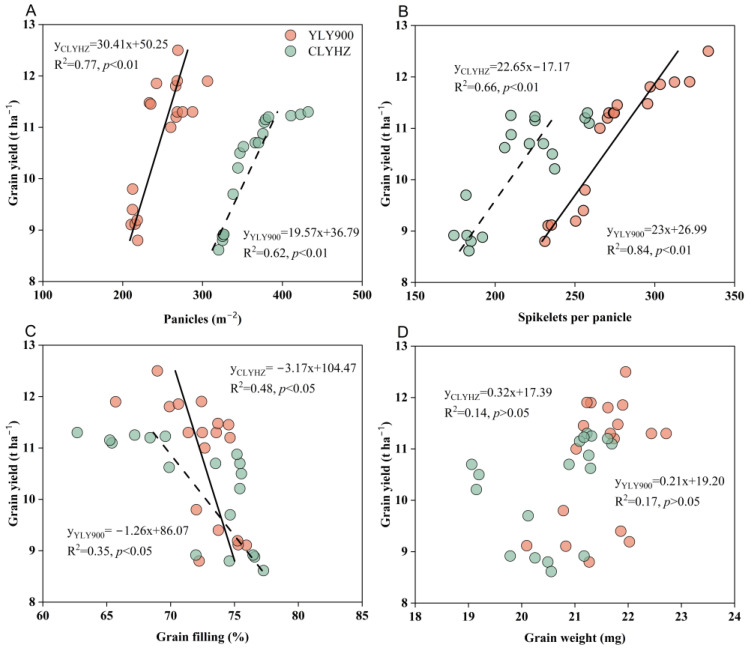
Relationships between grain yield and panicle number (**A**), spikelets per panicle (**B**), grain filling (**C**), and grain weight (**D**) in 2020 and 2021.

**Figure 6 plants-12-04063-f006:**
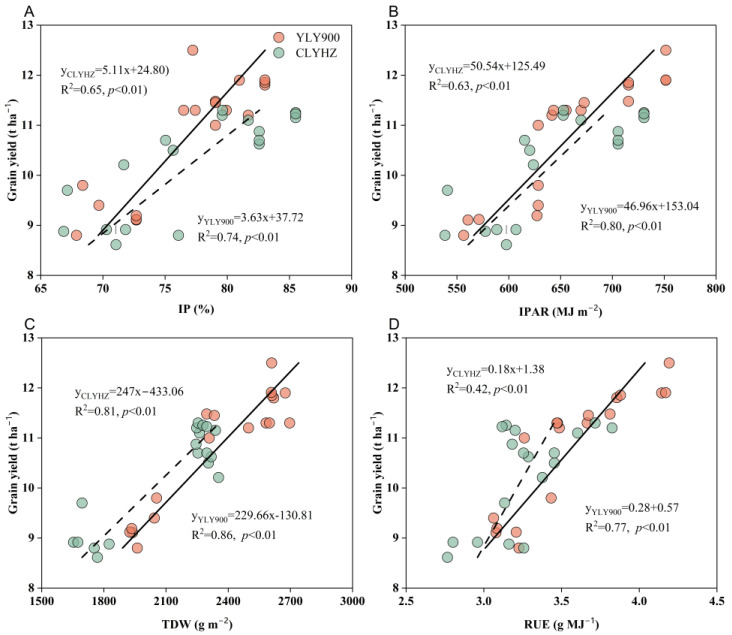
Relationship between grain yield and IP (**A**), IPAR (**B**), TDW (**C**), and RUE (**D**) in 2020 and 2021.

**Table 1 plants-12-04063-t001:** Analysis of variance (ANOVA) of the F-values of grain yield (GY), P, SP, GF, GW, IP, IPAR, TDW, and RUE at mature stage.

ANOVA	GY	P	SP	GF	GW	IP	IPAR	TDW	RUE
Year (Y)	0.50 ns	3.77 ns	1.04 ns	0.16 ns	6.81 *	0.73 ns	20.60 **	333.82 **	87.57 **
Nitrogen (N)	109.92 **	3.73 *	4.05 *	4.1 *	0.99 ns	254.55 **	363.82 **	766.94 **	55.33 **
Variety (V)	12.84 **	121.14 **	14.79 **	9.64 **	4.02 ns	130.91 **	404.31 **	25.10 **	223.36 **
Y × N	0.17 ns	0.18 ns	0.68 ns	0.11 ns	2.59 *	4.14 *	1.46 ns	20.68 **	11.70 **
Y × V	0.06 ns	0.03 ns	0.91 ns	3.65 ns	0.01 ns	6.02 *	1.49 ns	18.92 **	14.22 **
N × V	1.09 ns	2.10 ns	5.68 **	0.51 ns	0.15 ns	1.36 ns	6.00 **	7.79 **	2.58 ns
Y × N × V	0.61 ns	2.45 ns	8.13 **	7.42 **	0.17 ns	9.11 **	4.34 *	4.86 *	0.49 ns

GY: grain yield, P: panicles, SP: spikelets per panicle, GF: grain-filling, GW: grain weight, IP: light interception percentage, IPAR: intercepted photosynthetically active radiation; TDW: aboveground total dry weight; RUE: radiation use efficiency; ns: non-significant. ** Significant at *p* < 0.01. * Significant at *p* < 0.05.

**Table 2 plants-12-04063-t002:** Yield components of two panicle-type varieties under three N treatments at Jingzhou in 2020 and 2021.

Year	Nitrogen	Variety	P	SP	GF	GW
(Y)	(N)	(V)	(m^−2^)		(%)	(mg)
2020	N1	YLY900	214.5 b	270.0 a	72.3 b	21.3 a
		CLYHZ	326.2 a	186.3 b	75.3 a	20.3 b
	Mean		270.3 C	228.1 C	63.8 C	20.8 B
	N2	YLY900	268.1 b	298.7 a	73.6 b	22.0 a
		CLYHZ	379.6 a	234.5 b	75.5 a	19.1 b
	Mean		323.9 B	266.6 A	74.5 A	20.6 B
	N3	YLY900	289.6 b	322.6 a	68.7 a	21.9 a
		CLYHZ	431.9 a	257.7 b	65.5 b	21.5 a
	Mean		360.8 A	240.1 B	67.1 B	21.7 A
2021	N1	YLY900	214.8 b	233.2 a	75.5 b	21.0 a
		CLYHZ	324.0 a	180.1 b	77.5 a	20.5 a
	Mean		269.4 C	206.6 C	65.0 C	20.7 B
	N2	YLY900	236.9 b	254.0 a	73.3 a	21.3 a
		CLYHZ	347.4 a	208.6 b	72.5 a	21.1 a
	Mean		292.2 B	231.3 B	72.9 A	21.2 A
	N3	YLY900	264.9 b	274.2 b	71.0 a	21.6 a
		CLYHZ	370.6 a	223.8 a	67.3 b	21.2 a
	Mean		317.8 A	249.0 A	69.2 B	21.4 A

P: panicles; SP: spikelets per panicle; GF: grain-filling; GW: grain weight. Means with identical letters within each column do not exhibit statistically significant differences according to LSD (0.05). Distinct capital letters in the table indicate statistical significance at *p* < 0.01, whereas distinct lowercase letters indicate significance at *p* < 0.05.

**Table 3 plants-12-04063-t003:** Radiation use efficiency (RUE) and its related parameters of two hybrid rice varieties at maturity under different nitrogen rates in 2020 and 2021.

Year	Nitrogen	Variety	GD	IR	IP	IPAR	TDW at PH	TDW	RUE
(Y)	(N)	(V)	(d)	(MJ m^−2^)	(%)	(MJ m^−2^)	(g m^−2^)	(g m^−2^)	(g MJ^−1^)
2020	N1	YLY900	111	1639.8	68.6 a	562.8 a	1166.0 a	2019.7 a	3.6 a
		CLYHZ	108	1611.0	68.6 a	552.2 a	1036.3 b	1757.9 b	3.2 b
	Mean		110	1625.4	68.6 B	557.5 C	1101.2 C	1888.8 B	3.4 B
	N2	YLY900	111	1639.8	79.4 a	650.7 a	1444.0 a	2559.7 a	3.9 a
		CLYHZ	110	1639.8	75.6 b	619.7 b	1191.2 b	2301.9 b	3.7 a
	Mean		111	1639.8	77.5 A	635.2 B	1317.6 B	2430.8 A	3.8 A
	N3	YLY900	114	1661.6	78.5 a	652.5 a	1545.3 a	2660.4 a	4.1 a
		CLYHZ	110	1639.8	80.3 a	658.4 a	1214.2 b	2256.5 b	3.4 b
	Mean		112	1650.7	79.4 A	655.4 A	1379.7 A	2458.5 A	3.8 A
2021	N1	YLY900	102	1729.4	72.7 a	628.4 a	1085.1 a	1932.5 a	3.1 a
		CLYHZ	99	1682.5	71.0 a	597.7 b	873.8 b	1698.8 b	2.8 b
	Mean		101	1705.9	71.9 C	613.1 C	979.4 C	1815.7 C	3.0 C
	N2	YLY900	102	1810.4	79.0 b	715.5 a	1220.1 a	2312.4 a	3.2 a
		CLYHZ	99	1708.7	82.6 a	705.4 a	1196.2 a	2286.4 a	3.2 a
	Mean		101	1759.6	80.8 B	710.5 B	1208.1 B	2299.4 B	3.2 B
	N3	YLY900	105	1810.4	83.0 b	751.5 a	1438.0 a	2612.8 a	3.5 a
		CLYHZ	101	1708.7	85.5 a	730.4 b	1358.3 b	2304.1 b	3.2 b
	Mean		103	1759.6	84.3 A	741.0 A	1398.1 A	2458.5 A	3.4 A

GD: growth duration, IR: total solar radiation, IPAR: intercepted photosynthetically active radiation, IP: light interception percentage, TDW at PH: aboveground total dry weight at post-heading stage, TDW: aboveground total dry weight at maturity stage, RUE: radiation use efficiency. Means with identical letters within each column do not exhibit statistically significant differences according to LSD (0.05). Distinct capital letters in the table indicate statistical significance at *p* < 0.01, whereas distinct lowercase letters indicate significance at *p* < 0.05.

## Data Availability

All data supporting the conclusions of this manuscript are provided within the manuscript.

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
