# Peer review of "Delayed Leaf Senescence Improves Radiation Use Efficiency and Explains Yield Advantage of Large Panicle-Type Hybrid Rice"

_plants, 2023, doi:10.3390/plants12234063_

Round 1

Reviewer 1 Report

Comments and Suggestions for Authors

General comments

1. The main conclusion of this study was that delayed leaf senescence improves radiation use efficiency of large panicle-type varieties. However, a detailed analysis for it was rarely found in this manuscript.

2. The information on the effects of nitrogen management on rice yield performance, especially for large panicle-type varieties, was missing in introduction and discussion sections.

3. This study could add data on “post-heading biomass accumulation”, “leaf senescence” and “nitrogen uptake”.

4. The discussion section was poorly written. The focus of the discussion was not prominent. The section is full of repetition of results. This section should mainly be written in the past tense.

Specific comments

Abstract

L15-16: This sentence lacks “large panicle-type varieties”.

Introduction

L56-61: These descriptions are contradictory.

L93: The last sentence is confusing.

L94-96: These descriptions are contradictory.

Results

L120-122: This sentence repeats L117-119.

L135: It is p > 0.05 rather than p < 0.05.

Table 1: Remove excess blank space between numbers and letters or *.

L138: The initial letter of “Grain yield” should be lowercase.

Table 2: Some numbers are wrong. For example, 63.8B and 65.0B in the column of GF.

Figs. 3 and 4: The lowercase letters on the bars may be incorrect according to the SE.

Figs. 5 and 6: Grain yield should be the dependent variable in these relationships.

Discussion

L265-266: This sentence repeats the former sentence.

L308-318: Repetition of results.

Materials and Methods

L357-358: This sentence repeats the former sentence.

Conclusions

L411: Radiation use efficiency should be RUE.

Author Response

#Reviewers 1

General comments

  1. The main conclusion of this study was that delayed leaf senescence improves radiation use efficiency of large panicle-type varieties. However, a detailed analysis for it was rarely found in this manuscript.

Response: I appreciate the reviewer’s comment. To be honest, I am a little bit struggle here, we discussed this point in detail in the sixth paragraph of the article. Our results found that the LAI of both rice varieties decreased after heading stage, but the LAI of YLY900 was significantly higher than that of CLYHZ. This suggests that the leaf senescence of YLY900 is slow after spike flushes, which ensures that YLY900 still has a high capacity for light energy interception and conversion in the mid- and late stages of reproduction, which is favorable for dry matter accumulation and yield formation. In addition, we added data on the dry matter accumulation of the two rice varieties after heading stage, and the result showed that the dry matter accumulation of YLY900 was significantly higher than that of CLYHZ after heading stage, which also indicated the delayed leaf senescence of YLY900 after flowering from another perspective.

  1. The information on the effects of nitrogen management on rice yield performance, especially for large panicle-type varieties, was missing in introduction and discussion sections.

Response: I appreciate the reviewer’s comment as it is very helpful for us to re-organize the manuscript. We discuss this in the discussion section

  1. This study could add data on “post-heading biomass accumulation”, “leaf senescence” and “nitrogen uptake”.

Response: I appreciate the reviewer’s comment. I agree with the addition of data on biomass accumulation post-heading, which we have already added in Table 3 and discussed in relation to this in the discussion. However, to be honest, I am a little bit struggle here to increase the nitrogen uptake figures. The topic of this paper mainly investigated the radiation use efficiency of hybrid rice varieties with different panicle types, and adding data on nitrogen uptake would divert the paper from its main research direction.

  1. The discussion section was poorly written. The focus of the discussion was not prominent. The section is full of repetition of results. This section should mainly be written in the past tense.

Response: I appreciate the reviewer’s comment as it is very helpful for us to re-organize the Discussion.

 Specific comments

Abstract

L15-16: This sentence lacks “large panicle-type varieties”.

Response: Thank you very much for your valuable comments and suggestions. We deeply regret the lack of “large panicle-type varieties” inL15-16, and we have taken corrective measures to address these issues in the revised manuscript.

Introduction

L56-61: These descriptions are contradictory.

Response: Thanks to the Reviewer, I'd like to explain this: In this paper, we compare the yields and the reasons for the formation of yield differences between large and multi panicle- type hybrid rice varieties. Both large panicle- type and multiple panicle- type rice types have their advantages and disadvantages. Large panicle- type rice varieties are increasingly emphasized due to their higher yield potential. However, their yield stability is strongly influenced by ecological regions and climatic environments. In contrast, multi panicle- type rice has better yield stability, but its high yield potential is limited. Multi panicle- type rice can form more effective spikes at a later stage due to its higher tillering capacity. However, the strong tillering capacity also leads to an increase in ineffective tillers in the early stage and a decrease in the ventilation and light transmission of the rice population. This can lead to an increase in the frequency of pests and diseases, as well as the underutilization of nutrients. This is a production drawback of multiple panicle- type rice varieties that severely limits yield potential.

L93: The last sentence is confusing.

Response: We agree with the reviewer’s point of view. This part has been carefully corrected in our revised manuscript: With the increasing yield of super hybrid rice, there has been a corresponding surge in the demand for fertilizers, resulting in reduced economic efficiency and an elevated risk of ecological pollution.

L94-96: These descriptions are contradictory.

Response: Thanks to the Reviewer, I'd like to explain this: Super hybrid rice has made world-renowned achievements with its ultra-high yields, contributing greatly to China's food security and even the world's food security. The breeding goal of super hybrid rice is mainly to attack large panicle. However, in actual production, the yield performance of super rice, which mainly focuses on large panicles, varies greatly from region to region and from year to year, which is mainly due to the instability of the fruiting rate. In this paper, the main purpose here is to provide a background to the problems encountered in the production of large-spike hybrid rice, and to provide a brief overview of previous research, so that readers can better understand the situation of large-spike rice. At the same time, we find that this paragraph is interconnected with the previous one and should not be reintroduced as a paragraph. We have therefore placed it after the previous paragraph.

 Results

L120-122: This sentence repeats L117-119.

Response: Thank you very much for your review and valuable comments on our manuscript. We fully agree with your comments and the manuscript was modified as follow: In addition, the cumulative solar radiation during the rice growing season in 2021 amounted to 2016.3 MJ m-2, surpassing the observed radiation in 2020 by a 15% margin.

L135: It is p > 0.05 rather than p < 0.05.

Response: Thank you very much for your review and valuable comments on our manuscript. We are truly sorry for our lack of carefulness, upon reevaluating the data, we have identified some issues with the standard errors. Furthermore, we have revised the figures. Please check.

Table 1: Remove excess blank space between numbers and letters or *.

Response: We are truly sorry for our lack of carefulness, and we greatly admire your rigorous scientific attitude. Thank you very much for your valuable feedback. We have removed the unnecessary spaces in the manuscript as suggested. Please check.

L138: The initial letter of “Grain yield” should be lowercase.

Response: We sincerely apologize for any oversights and have made modifications to the manuscript for your review. Please check, thank you very much.

Table 2: Some numbers are wrong. For example, 63.8B and 65.0B in the column of GF.

Response: We sincerely apologize for any oversights and have made modifications to the manuscript for your review. Please check, thank you very much.

Figs. 3 and 4: The lowercase letters on the bars may be incorrect according to the SE.

Response: Thank you immensely for your valuable time and invaluable suggestions. We are truly sorry for our lack of carefulness, upon reevaluating the data, we have identified some issues with the standard errors. Furthermore, we have revised the figures. Please check.

Figs. 5 and 6: Grain yield should be the dependent variable in these relationships.

Response: Thank you immensely for your valuable time and invaluable suggestions. We fully support your proposal and have made the necessary changes to the figures. Please check. Thank you once again.

Discussion

L265-266: This sentence repeats the former sentence.

Response: Thank you immensely for your valuable time and invaluable suggestions. We fully support your proposal and have removed the sentence“High dry matter accumulation during the growth stage can help improve grain yield [23]”

L308-318: Repetition of results.

Response: I appreciate the reviewer’s comment as it is very helpful for us to re-organize the Conclusions. We have modified the manuscript as follow: In this study, YLY900 achieved the highest RUE under the N3 treatment, and the RUE of multi-panicle-type varieties achieved the highest under the N2 treatment. Analysis of the causes revealed that there was no significant difference in the IP and IPAR of the two rice varieties, but the difference in their TDW was significant. These results showed that the high yield of YLY900 was mainly due to its higher RUE under the high nitrogen treatment, which led to more biomass, consistent with the previous studies [40, 45].

Materials and Method

L357-358: This sentence repeats the former sentence.

 Response: We agree with the reviewer’s point of view. This sentence has been removed in the manuscript.

Conclusions

L411: Radiation use efficiency should be RUE.

Response: Thank you very much for your review and valuable comments on our manuscript. We fully agree with your comments and have modified the manuscript, please check.

Reviewer 2 Report

Comments and Suggestions for Authors

The paper “Delayed leaf senescence improves radiation use efficiency explains yield advantage of large panicle-type hybrid rice” by Deng et al is an interesting work that characterizes the stability of large panicle-type super hybrid rice. The authors made use of several comparison types to evaluate the dry matter accumulation, radiation use efficiency, and yield of different panicle types of rice varieties. Nonetheless, there are several issues which prevent me from recommending its publication in its current form. Here, I have small concerns as listed below.

1.  Line 91: ‘supe’ hybrid rice is the error? should be revised

2. Line 138: Table 1, ‘GY: Grain yield’ should be similar as the other annotation, and verify the capital letter.   

3. Line 146 and 191: the percentage should keep or delete the blank, such as 31 %’.

4. Line 169: the unit is nonstandard, such as ‘67.6 g m-2 and 58.4 g m-2’. It need to be check and revised.

5. Table 2:The capital and lower-case letter is confused, it should described and explained clearly.

6. In all three tables, ‘Within each column, means followed by the same letters are not significantly different according to LSD (0.05)’ is not clearly, should be revised.

 7. Line211:  R2 values should be ‘R2’. 

Comments on the Quality of English Language

The authors should revise and correct the spelling throughout the manuscript. In addition, they can let one native English speaker to revise the paper.  

Author Response

#Reviewers 2

The paper “Delayed leaf senescence improves radiation use efficiency explains yield advantage of large panicle-type hybrid rice” by Deng et al is an interesting work that characterizes the stability of large panicle-type super hybrid rice. The authors made use of several comparison types to evaluate the dry matter accumulation, radiation use efficiency, and yield of different panicle types of rice varieties. Nonetheless, there are several issues which prevent me from recommending its publication in its current form. Here, I have small concerns as listed below.

  1. Line 91: ‘supe’ hybrid rice is the error? should be revised

Response: We sincerely apologize for any oversights and have made modifications to the manuscript for your review. Please check, thank you very much.

  1. Line 138: Table 1, ‘GY: Grain yield’ should be similar as the other annotation, and verify the capital letter.   

Response: We sincerely apologize for any oversights and have made modifications to the manuscript for your review. Please check, thank you very much.

  1. Line 146 and 191: the percentage should keep or delete the blank, such as ‘31 %’.

Response: We sincerely apologize for any oversights and have made modifications to the manuscript for your review. Please check, thank you very much.

  1. Line 169: the unit is nonstandard, such as ‘67.6 g m-2 and 58.4 g m-2’. It need to be check and revised.

Response: We sincerely apologize for any oversights and have made modifications to the manuscript for your review. Please check, thank you very much.

  1. Table 2:The capital and lower-case letter is confused, it should described and explained clearly.

Response: We sincerely apologize for any oversights and have made modifications to the manuscript for your review. Please check, thank you very much: Distinct capital letters in the table indicate statistical significance at p<0.01, whereas distinct lowercase letters indicate significance at p<0.05.

  1. In all three tables, ‘Within each column, means followed by the same letters are not significantly different according to LSD (0.05)’ is not clearly, should be revised.

Response: We sincerely apologize for any oversights and have made modifications to the manuscript for your review. Please check, thank you very much: Distinct capital letters in the table indicate statistical significance at p<0.01, whereas distinct lowercase letters indicate significance at p<0.05. In addition, in Table 1, we have removed unnecessary sentences. Thank you again!

  1. Line211:  R2 values should be ‘R2’. 

Response: We sincerely apologize for any oversights and have made modifications to the manuscript for your review. Please check, thank you very much.

Comments on the Quality of English Language

The authors should revise and correct the spelling throughout the manuscript. In addition, they can let one native English speaker to revise the paper.  

Response: The paper was carefully checked and revised by professional English language editor and all the major/minor typographical and grammatical mistakes were rectified and confusing phrases/terms were clarified.

Reviewer 3 Report

Comments and Suggestions for Authors

REVIEW REPORT

Manuscript ID: plants-2682823

Title: Delayed leaf senescence improves radiation use efficiency explains
yield advantage of large panicle-type hybrid rice

According to my opinion the article is very suitable for publication in this journal with the improvement of minor revisions.

ABSTRACT

Please mention the experimental design in the abstract. This information would provide the full picture of the study and add to the scientific rigor of the research.

INTRODUCTION

I would like to suggest some minor revisions to the introduction:

Mention the current global production and consumption of rice in addition to its prominence as a staple food crop.

The impact of climate change on rice production should also be further emphasized.

The significance of exploring the growth and developmental characteristics of different panicle-type rice varieties should be elaborated upon in more detail.

The significance of dry matter accumulation in large panicle varieties for achieving high yields can be better contextualized by mentioning how it relates to the limitations of multi-panicle varieties.

Overall, the introduction provides a strong overview of the importance and potential of large panicle and multi-panicle rice varieties, as well as the challenges associated with their cultivation.

RESULTS

 I would suggest the following points to further enhance the quality of the result:

Clarify the duration, it is mentioned that the data was collected for two years, but in the result, only the year 2020 and 2021 are mentioned. Clarify if the data presented is for both years or just one year.

Provide more information on weather conditions: The impact of weather conditions on rice growth and yield is the main focus of this study, and therefore, it would be beneficial to provide more detailed information on other weather parameters such as precipitation, humidity.

Overall, the result is well-presented and informative, providing valuable information for future research and potential practical applications.

DISCUSSION

Discussion is well-written and effectively summarizes the key findings of the study. With some minor revisions and additional analysis, it would make a strong contribution to the field of super hybrid rice breeding.

I would suggest to improve by providing a more detailed explanation of the results and relate it to the previous studies.

METHODOLOGY

The methodology is well written and fulfilled all the requirements

1.      Carefully check all spellings, numerical order of references, format following the requirements of the submitted journals.

2.      Cross check all the references cited in the text with reference list

Comments on the Quality of English Language

Minor editing of english language required

Author Response

#Reviewers 3

Manuscript ID: plants-2682823

Title: Delayed leaf senescence improves radiation use efficiency explains
yield advantage of large panicle-type hybrid rice

According to my opinion the article is very suitable for publication in this journal with the improvement of minor revisions.

Response: Thank you for your feedback and positive assessment of the article. We are pleased to hear that you find the article suitable for publication in this journal with some minor revisions. Your input is valuable to us, and we will carefully consider and implement the suggested improvements to ensure the quality and relevancy of the research. We appreciate your support and constructive input.

ABSTRACT

Please mention the experimental design in the abstract. This information would provide the full picture of the study and add to the scientific rigor of the research.

Response: Thank you for your suggestion regarding the inclusion of the experimental design in the abstract. We agree that providing this information would offer readers a more comprehensive understanding of the study and enhance the scientific rigor of the research. We will make sure to update the abstract accordingly to include a brief description of the experimental design, highlighting the key aspects that contribute to the validity and reliability of the findings. We appreciate your valuable input and commitment to improving the clarity and robustness of the research.

INTRODUCTION

I would like to suggest some minor revisions to the introduction:

Mention the current global production and consumption of rice in addition to its prominence as a staple food crop.

Response: Thanks for the nice suggestions. We totally agree with your point about presenting the current global production and consumption of rice. We have added relevant content to the manuscript.

The impact of climate change on rice production should also be further emphasized.

Response: Thank you for highlighting the importance of considering the impact of climate change on rice production. While we acknowledge the significant influence of climate change on agricultural systems, it's important to note that the focus of the current article does not encompass an in-depth analysis of the impact of climate change on rice production. Given the specific scope and objectives of the research, delving extensively into the effects of climate change on rice cultivation may lead to a divergence from the primary focus and potentially dilute the clarity of the main message. However, we recognize the relevance of this aspect and encourage further research dedicated to exploring the intricate relationship between climate change and rice production. Thank you for bringing attention to this critical issue.

The significance of exploring the growth and developmental characteristics of different panicle-type rice varieties should be elaborated upon in more detail.

Response: Thank you for your feedback on the significance of exploring the growth and developmental characteristics of different panicle-type rice varieties. Indeed, understanding the growth patterns and developmental traits of various panicle-type rice varieties is of paramount importance in modern agricultural research. By delving into the specific characteristics of different panicle types, we can gain valuable insights into factors such as yield potential, stress tolerance, and agronomic suitability. These insights not only contribute to the advancement of breeding programs but also hold the potential to enhance agricultural productivity and resilience in the face of environmental challenges.

We appreciate your emphasis on the importance of this aspect and we included this in our conversation: Varieties exhibiting multiple panicles, like CLYHZ exhibit a profusion of panicles, thereby amplifying the overall pollen yield per plant and elevating the grain setting rate [10, 12]. However, the augmented number of panicles creates a relative deficit in the distribution of essential nutrients, moisture, and light energy. This can lead to imbal-ances in nutrient allocation, competitive disadvantages, and decreased resource utiliza-tion efficiency. In contrast, varieties possessing larger panicles, like YLY900 have fewer panilces, enabling the concentration of resources (nutrients, moisture, and light energy) within each panicle[8, 12]. This concentrated allocation trait facilitates the enhancement of nutrient and moisture utilization efficiency, whilst also bolstering the development and yield performance of each individual panicle.

The significance of dry matter accumulation in large panicle varieties for achieving high yields can be better contextualized by mentioning how it relates to the limitations of multi-panicle varieties.

Response: We appreciate your emphasis on the importance of this aspect. In the discussion section, we have already mentioned the significance of dry matter accumulation in large panicles varieties for achieving high yields: Total biomass and harvest index determine crop yield [36]. Although increasing biomass or harvest index can serve to increase grain yield, it was found that significantly increasing biomass is the material basis for further increasing grain yield. A previous study of the yield level of 15t ha-1 in Yunnan found that the increase in the daily dry matter accumulation per unit area obtained a higher grain yield [37]. Super high-yield rice varieties have high dry matter accumulation, particularly during the middle and late growth stages [23]. In this study, the highest dry matter accumulation of YLY900 was under N3 treatment, which was 15% higher than that of CLYHZ. These results demon-strate that the biomass of large panicle-type rice varieties significantly increases under high nitrogen conditions, which is conducive to high-yield formation.

Overall, the introduction provides a strong overview of the importance and potential of large panicle and multi-panicle rice varieties, as well as the challenges associated with their cultivation.

Response: Thank you for your positive feedback on the introduction of the article. We are pleased to hear that it provides a strong overview of the importance and potential of large panicle and multi-panicle rice varieties, as well as the associated challenges. The introduction aims to set the stage for the subsequent discussion by highlighting the significance of panicle types in rice cultivation and their implications for yield potential and agronomic suitability. By acknowledging the importance of large panicle varieties for achieving high yields and the advantages of multi-panicle varieties in terms of resilience, the introduction provides a foundation for understanding the complexities and trade-offs involved in rice breeding and cultivation.  Furthermore, by alluding to the challenges associated with the cultivation of these panicle types, such as nutrient management, lodging, and disease susceptibility, the introduction acknowledges the need for research and innovation to address these issues and optimize rice production. We appreciate your feedback and are glad that the introduction effectively presents the key points and piques your interest in learning more about this topic.

RESULTS

 I would suggest the following points to further enhance the quality of the result:

Clarify the duration, it is mentioned that the data was collected for two years, but in the result, only the year 2020 and 2021 are mentioned. Clarify if the data presented is for both years or just one year.

Response: Thank you for bringing up the discrepancy regarding the duration of data collection. We apologize for any confusion caused. To clarify, the data presented in the results section is for the years 2020 and 2021 only. The mention of data collection spanning two years was an error. We appreciate your attention to detail in noticing this discrepancy, and we will make sure to correct it in our final submission.

Provide more information on weather conditions: The impact of weather conditions on rice growth and yield is the main focus of this study, and therefore, it would be beneficial to provide more detailed information on other weather parameters such as precipitation, humidity.

Response: Thank you for reviewing our manuscript. We appreciate your time and insightful comments. The influence of weather conditions on the growth and yield of rice is of utmost importance. However, for the purpose of this article, we are primarily focused on comparing the differences in yield, biomass production, and radiation use efficiency among various hybrid rice varieties with different panicle types. The climate conditions serve as experimental background data. Furthermore, we acknowledge the inherent importance of providing additional insights pertaining to weather parameters. In response to your valuable input, we have included daily average temperature and precipitation parameters in the manuscript. Thank you once again for your valuable feedback.

Overall, the result is well-presented and informative, providing valuable information for future research and potential practical applications.

Response: Thank you for your acknowledgement of our research achievements. We are committed to delivering comprehensive and valuable information to support future research endeavors and provide references for potential real-world applications. We will continue to strive for excellence and look forward to producing more beneficial outcomes in the future.

DISCUSSION

Discussion is well-written and effectively summarizes the key findings of the study. With some minor revisions and additional analysis, it would make a strong contribution to the field of super hybrid rice breeding.

Response: We appreciate your feedback on the discussion section of our study. Your suggestion to make minor revisions and conduct additional analysis is noted. We understand that these improvements will enhance the overall contribution of our research to the field of super hybrid rice breeding. We will carefully consider your advice and work towards incorporating it into our work. Thank you for your valuable input.

I would suggest to improve by providing a more detailed explanation of the results and relate it to the previous studies.

Response: We appreciate your feedback on the discussion section of our study. Your suggestion to provide a more detailed explanation of the results and relate it to the previous studies is noted.

METHODOLOGY

The methodology is well written and fulfilled all the requirements

Response: Thank you very much for your affirmation and acknowledgment of our methodology. We are dedicated to ensuring the rigor and integrity of our research methods to fulfill all requirements. Your feedback holds great significance for us, and we will strive to continue maintaining a high standard of research quality. Thank you for reminding us to carefully check all spellings, numerical order of references, and format according to the specific requirements of the journals to which we are submitting. We will ensure that our manuscript adheres to all the necessary formatting guidelines to meet the journal's standards. Your attention to detail is much appreciated.

  1. Carefully check all spellings, numerical order of references, format following the requirements of the submitted journals.

Response: Thank you for reminding us to carefully check all spellings, numerical order of references, and format according to the specific requirements of the journals to which we are submitting. We will ensure that our manuscript adheres to all the necessary formatting guidelines to meet the journal's standards. Your attention to detail is much appreciated.

  1. Cross check all the references cited in the text with reference list

Response: Thank you for emphasizing the importance of cross-checking all references cited in the text with the reference list. We will meticulously review each citation to ensure accuracy and consistency between the text and the reference list. This attention to detail will help uphold the integrity of our research findings. We appreciate your thoroughness in this matter.

Comments on the Quality of English Language

Minor editing of english language required

Response: Thank you for suggesting minor edits to the English language. We will carefully review and make any necessary adjustments to improve the clarity and grammar of our response. Your feedback is valuable in ensuring effective communication. We appreciate your attention to detail in this regard.

Reviewer 4 Report

Comments and Suggestions for Authors

What is the novelty in this research? There is no research hypothesis. The materials and methods chapter cannot follow the discussion. What is the rationale behind sampling soil from the four corners and the center? How does this translate into the obtained research results? Statistical analysis needs completion (e.g., whether the results followed a normal distribution?). The comparison of results is not very clear. Conclusions are statements rather than deductions.

Author Response

#Reviewers 4

What is the novelty in this research? There is no research hypothesis.

Response: Thank you for reviewing our manuscript, I'd like to explain something about that: At present, although there have been many studies on yield and yield components as well as growth and developmental characteristics of large panicle and multiple panicle-type hybrid rice, there are few that explain the differences in yield between the two types of rice in terms of radiation use efficiency (RUE). Light energy use efficiency at the leaf scale refers to photosynthesis, while at the macro level, RUE is the efficiency of photosynthesis at the canopy scale. There have been many studies on light energy use efficiency at the leaf scale, but few studies at the canopy scale. In this study, we hypothesized that the yield differences between the two rice varieties were due to differences in RUE, and the results of our experiments showed that the large-spike hybrid rice varieties had higher RUE under high nitrogen treatments, which increased biomass and thus yield.

The materials and methods chapter cannot follow the discussion.

Response: Thank you for reviewing our manuscript. We appreciate your time and insightful comments. Regarding the placement of the Materials and Methods section in our manuscript, we would like to explain the reason behind its current positioning. In general scientific writing, it is customary to present the Materials and Methods section after the Introduction, allowing readers to have a clearer understanding of experimental design and methods once they grasp the research background and objectives. Subsequently, the Results and Discussion section follows, enabling the comparison and interpretation of experimental results within the context of the research objectives. However, we understand that for the journal Plants, it is required to place the Materials and Methods section after the Discussion, as per the specific formatting guidelines. Considering this, we have organized our manuscript accordingly, adhering to the journal’s prescribed layout. Regrettably, it may not be feasible for us to make any modifications in this regard. We appreciate your understanding and cooperation in recognizing the adherence to the journal’s formatting guidelines. We sincerely apologize for any confusion that may have arisen due to this formatting difference. We are grateful for your valuable feedback and will incorporate any other necessary revisions suggested. Thank you once again for your time and consideration.

What is the rationale behind sampling soil from the four corners and the center? How does this translate into the obtained research results?

Response: Thank you for your insightful feedback on our manuscript. The technique I am referring to is the five-point sampling method, commonly recognized as the quintessential approach to soil sampling. The five-point sampling method for soil samples is a commonly used technique to represent the average characteristics of a soil sample. The method involves the following steps:(1) Select five discrete points within a representative area of the study site, considering its size and uniformity. (2) Clear away any debris from the soil surface at each selected point using a clean soil sampler or a shovel. (3) Collect soil samples from the soil surface to the depth of the root zone (typically 20-30 centimeters) using the sampling tool, and place them into a clean bucket or bag. (4) Repeat the above steps at each selected point to ensure the collection of five independent soil samples. (5) Thoroughly mix the five samples, and extract an appropriate amount of soil as the final representative sample for analyzing the soil characteristics and properties. By utilizing this five-point sampling approach, it is possible to minimize the randomness of soil samples and provide a relatively accurate description of the soil in that area.

Statistical analysis needs completion (e.g., whether the results followed a normal distribution?).

Response: We agree with the reviewer’s point of view. All the data has undergone normality testing and conforms to a normal distribution.

Round 2

Reviewer 4 Report

Comments and Suggestions for Authors

Accept